# Viral Zoonotic Diseases of Public Health Importance and Their Effect on Male Reproduction

**Olabisi Lateef Okeleji** [1], **Lydia Oluwatoyin Ajayi** [2], **Aduragbemi Noah Odeyemi** [3], **Victor Amos** [3], **Hezekiah Oluwatobi Ajayi** [3], **Amos Olalekan Akinyemi** [2], **Chibueze Samuel Nzekwe** [4], **Johnson Wale Adeyemi** [5] and **Ayodeji Folorunsho Ajayi** [3,*]

1   Obafemi Awolowo University Teaching Hospital, Ile-Ife PMB 5538, Osun State, Nigeria
2   Department of Biochemistry, Ladoke Akintola University of Technology, Ogbomoso PMB 4000, Oyo State, Nigeria
3   Department of Physiology, Ladoke Akintola University of Technology, Ogbomoso PMB 4000, Oyo State, Nigeria
4   Department of Biochemistry, Adeleke University, Ede PMB 250, Osun State, Nigeria
5   Department of Physiology, Adeleke University, Ede PMB 250, Osun State, Nigeria
*   Correspondence: aajayi22@lautech.edu.ng; Tel.: +234-80-3383-4495

**Simple Summary:** There exists an almost unavoidable relationship between humans and animals. Humans keep animals as pets and use some for food. However, one should note that disease-causing organisms such as viruses can be present in these animals and be transmitted to humans, thereby causing life-threatening sickness or negatively impacting fertility. This article presents the existing evidence on the effects of some communicable viral diseases on male fertility. Knowing that infertility is a major concern to couples and families and that viral diseases affect men of all ages and locations, the authors question whether or not these viral diseases contribute to the state of infertility in men. This will help men and the general populace understands the extent of the damage of viral diseases transmitted by animals to their reproductive function and possible infertility. We believe this review is of educational, scientific and societal value.

**Abstract:** Zoonotic diseases occur as a result of human interactions with animals with the inadvertent transmission of pathogens from one to another. Zoonoses remain a major cause of morbidity and mortality among human populations, as they have been a source of pandemics in human history. Viral zoonoses account for a significant percentage of pathogens of zoonotic sources, posing a huge risk to men's general health and fertility. This review identifies the existing knowledge on the effects of viral zoonotic diseases on male fertility. Evidence from reviewed articles showed that viral zoonotic diseases elicit an immune reaction that induces inflammatory mediators and impairs testicular functions such as spermatogenesis and steroidogenesis, leading to abnormal semen parameters that lead to subfertility/infertility. Although most zoonotic viruses linger in semen long after recovery, their presence in semen does not directly translate to sexual transmission. There is a need to further delineate the possible risk of the sexual transmission of these diseases. While a few of the viral zoonotic diseases discussed have been well-studied, there is a need to place attention on others so as to fully understand their effects on male reproduction and therefore take the right steps towards preserving male fertility.

**Keywords:** fertility; public health; spermatogenesis; steroidogenesis; zoonoses

## 1. Introduction

Zoonotic diseases are infections that occur as a result of human interactions with animals and the environment [1]. These diseases are transmissible from animals to humans or from humans to animals [2]. Human–animal interactions occur mostly through human

contact with animal-origin foods, domestic animals, pets, aquatic animals, edible insects and foodborne pathogens [3].

Animal-borne pathogens constitute about 61% of all human disease-causing organisms, and three out of every four new emerging infectious diseases arise from animal sources [2,4]. Zoonotic diseases disrupt human activities and cause increased morbidity and mortality among the human population, being responsible for about 2.4 billion cases of illness and about 2.7 million deaths in low- and middle-income countries [5]. The impact of zoonotic diseases on human health is evident from previous outbreaks that have plagued the human race on a global scale. To mention a few, the COVID-19 outbreak was first reported in Wuhan, China, and paralyzed human activities in 2020 with an ongoing negative effect on the world economy. Other notable zoonotic disease outbreaks include the Ebola virus outbreak in 2013, the Russian flu in 1977, the Spanish flu in 1918, the Middle East Respiratory Syndrome Coronavirus (MERS-CoV) in 2012, Human Immunodeficiency virus in 1981 and Swine flu in 2009 [6]. Exposure to zoonotic pathogens disrupts human physiology with short- or long-term effects on body organ functions.

Microbial infection from various sources is a major cause of infertility. Many microbial pathogens have been previously ascribed as a cause of different female reproductive dysfunctions, which might cause infertility or delayed fertility. Infection is considered to be a contributing factor in male infertility when it affects the urogenital tract and promotes inflammatory responses. Infectious agents disrupt the internal milieu of the male reproductive system, leading to numerous pathologies, such as orchitis, epididymitis, prostatitis and urethritis. Genital tract infection accounts for about 15% of male infertility cases by disrupting spermatogenesis at different stages [7,8]. Pathogens cause male reproductive dysfunction by inflicting direct harm on the reproductive tissues or activating an immunological response and the release of inflammatory cytokines, which leads to oxidative stress and consequent oxidative damage [9]. In addition, infections in the male reproductive tract may be detrimental to sperm quality [7] or cause testicular dysfunction through the disintegration of its physical barrier and leukocyte infiltration [10].

While the effects of infections on female reproduction have been well explored, the effects of viral zoonotic diseases on male reproduction have not been well characterized. Thus, the present review aims to explore the existing knowledge on the effects of viral zoonotic diseases on male reproductive fertility. While the global range of viral zoonotic diseases is large, this review focuses on the effects of influenza, COVID-19, Crimean–Congo virus, Ebola virus, Zika virus, Lassa fever and monkeypox virus on male fertility.

*Search Method*

Research-based databases such as Google scholar, PubMed and Web of Science were used for this review. Each of the viruses included in this study were reviewed by following a keyword format that juxtaposed the virus with (male fertility or testes or semen parameters or sperm parameters). For instance, the keyword syntax search for coronavirus was ((((COVID-19) OR (SARS-CoV-2)) OR (Coronavirus)) AND (((((Male fertility) OR (Testes)) OR (Semen parameters)) OR (Sperm parameters)). The articles included in this study are original research articles on humans and other mammals, while research on other animals and those not reported in English were excluded. Major attention was paid to articles published between 1990 and the present.

## 2. Effects of Viral Zoonotic Diseases on Male Reproduction

### 2.1. Effect of Influenza on Male Fertility

A number of studies have shown that influenza alters sperm parameters [11–13]. Interestingly, the virus has not been previously seen in semen, nor has the viral receptor been reportedly expressed in the human genital system [13]. However, experimental models studying the effect of human influenza on male fertility suggest that the virus can induce chromosomal aberrations in spermatozoa [14,15].

A study that evaluated the effects of an influenza outbreak on sperm production in Boaz reported that influenza caused a decrease in sperm production when compared to pre-outbreak values, which was later reversed after a period of observation [16]. Similarly, Sergerie et al. reported that patients with febrile episodes had abnormal semen parameters and sperm DNA integrity, which could result in future infertility [17]. Furthermore, Evenson et al. reported that influenza and the febrile condition could have latent effects on the sperm chromatin structure and might lead to the transient release of abnormal sperm [12].

One of the notable symptoms of influenza is fever. High fever may affect the chromatin structure, which leads to abnormal sperm cells. In addition, fever may necessitate the prescription of acetaminophen in influenza patients. Acetaminophen has been shown to suppress the synthesis of testosterone and produce oxidative stress, causing the death of sperm cells. In addition, pain medications have an effect on nitric oxide production, which plays a key role in sperm motility and in preserving normal sperm function. Furthermore, painkillers reduce the synthesis of prostaglandins, which in turn affects sperm motility [18].

A viral infection affects the testes because they are primarily blood-borne. Influenza may lead to orchitis and thereby impair testicular function [19]. Orchitis may lead to the impaired function of the testis and consequent male infertility. From the studies previously stated, the mechanisms suggested to impair the normal functioning of the testes during influenza infection are:

Elevated body temperature (fever), which damages testicular germ cells;

Inflammation of the testes (orchitis) with the associated disruption of the steroidogenic and spermatogenic functions of the testes [17];

Pyrogenic and hemagglutinating properties of influenza virus;

Potential indirect effects on testicular function through increased levels of inflammatory cytokines.

### 2.2. COVID-19 and Male Fertility

SARS-CoV-2 directly infects the cells by coupling with angiotensin-converting enzyme 2 (ACE-2) receptors and transmembrane serine protease 2 (TMPRSS2) [20]. ACE-2 is present on almost all testicular cells. The results of a genotype tissue expression project showed that the tissue expression of ACE2 is highest in the testicles, while TMPRSS2 is mostly expressed in the prostate [21]. A report from single-cell RNA sequencing data demonstrates that the testes lack the co-expression of ACE2/TMPRSS2 enzymes [22]. The lack of the co-expression of these two enzymes in the male gonads suggests that SARS-CoV-2 might not be able to infect gonad cells. However, there is high expression of ACE2 in the genito-urinary tissues, which could contribute to the infectivity of SARS-CoV-2 in the reproductive tissues. It has been previously reported that the SARS coronavirus could have an impact on male gonads. Hence, SARS-CoV-2 is likely to affect testicular tissue, semen parameters and male fertility [23].

Before the COVID-19 outbreak in December 2019, there were two previous outbreaks of coronaviruses, and since then, the possible involvement of the testes and genito-urinary tract has been studied [24]. A study involving six men who died of COVID-19 in China reported that these patients had orchitis, germ cell destruction, very low spermatozoa in seminiferous tubules, basement membrane thickening, leukocyte infiltration and vascular congestion, which suggests systemic complications [24]. Another study in Brazil on the first autopsies of COVID-19 patients reported that orchitis with fibrin microthrombi was a common feature in the post-mortem analysis of testicular samples. Similarly, a study of 12 deceased COVID-19 patients demonstrated a reduced Leydig cell count, mild infiltration of leukocytes and significant seminiferous cellular injury [25].

From the results of the studies conducted, one can suggest that SARS-CoV-2 reaches the testis through the blood and affects the Leydig and Sertoli cells, thereby altering the steroidogenic pathway and also recruiting immune cells, which might promote inflammatory markers and promote orchitis. One can also hypothesize that SARS-CoV-2 may

infect the testes directly, thereby inducing a cytokine storm, as is the case in other viral infections [26–28].

As regards the effects of COVID-19 infection on the endocrine function of the testis, a study compared the hormone levels of infected men with an age-matched control group. The study reported a significant increase in luteinizing hormone and a reduction in the serum testosterone/LH ratio in the COVID-19 group. The study also reported an inverse correlation between the testosterone/LH ratio and white blood cell counts and C-reactive protein, which suggests an immunological reaction to Leydig cells, which might lead to temporary hypogonadism [29]. A systematic review of the impact of COVID-19 on male fertility reported the absence of the virus in semen but noted that it could affect seminal parameters, induce orchitis and cause hypogonadism [30]. The possible risk factors of coronavirus disease 2019 (COVID-19) infection on fertility come from the abundance of angiotensin-converting enzyme-2 (ACE2) receptors on the testes, a reduction in important sex hormone ratios and COVID-19-associated fever [31].

On the contrary, the results of some studies suggest that the involvement of the testes in COVID-19 might be minimal. A study on patients with mild to severe symptoms of COVID-19 reported that six out of thirty-four men complained of some scrotal discomfort at diagnosis. In addition, on physical examination and Doppler study, orchitis was not confirmed in these patients [32]. A similar study reported that just 10% of the testes examined were found to contain SARS-CoV-2 on direct examination. Some other studies also reported that they did not establish the presence of SARS-CoV or SARS-CoV-2 in testicular biopsy samples [33]. This seeming inconsistency suggests the need for further studies to clarify and understand the effects of COVID-19 on male fertility.

### 2.3. Zika Virus and Infertility

Zika virus is primarily spread through mosquito bites (Aedes species), but it can also spread through sex. Zika can stay in the semen and may be passed to a partner (and the fetus) for months after infection, even if there are no symptoms [34]. Reports from recent studies show that Zika virus RNA persists in the semen and also in the male and female reproductive tracts, which suggests the possibility of sexual transmission [35]. The sexual transmission of Zika virus was first reported in 2011 and supported by many occurrences of many other cases [36]. In patients previously infected with Zika virus, viral RNA has been reported to persist in their semen six months post-infection [37]. This has been supported by previous studies that reported the presence of Zika virus RNA in symptomatic and previously infected asymptomatic patients [38]. Similarly, Zika virus has been shown to remain in the semen and sperm fraction needed for assisted reproduction for up to 112 days post-infection [39]. All of these findings provide evidence that men previously infected with Zika virus could be a reservoir of infection through sexual transmission [40].

An experimental model of the sexual transmission of Zika virus reported that Zika virus RNA is mainly shed from the leukocytes and epithelial cells of the epididymis [41]. In the initial stages of Zika infection, it represses cell growth and proliferation and also disrupts cell-to-cell signaling between germ cells and Sertoli cells [42]. Further, it suppresses the ability of Sertoli cells to secrete inhibin B [43]. As Zika infection persists, it drives an immune response in the Sertoli cells by upregulating pro-inflammatory genes such as human leukocyte antigen class I (HLA), interleukin-23 subunit alpha, interleukin 6 and lymphotoxin beta (LTB) [41].

The innate immune system mainly contributes to Zika virus spread in the testicles through the response of interferons. A study on the effect of Zika virus in interferon receptor knockout mice shows that early in the infection phase, Zika virus does not induce the interferon type 1 receptor but presents a modest induction after 48 to 72 h. After 72 h, the level of pro-inflammatory cytokines was noted to be significantly increased in samples taken from infected Sertoli cells. Despite the robust immune response from Sertoli cells, Zika virus lingers in the reproductive tract of males long after infection [44].

The ability of ZIKV to infect the male genital tract, especially the testis, and result in its sexual transmission is because the ZIKV cofactor/attachment receptor AXL is differentially expressed in the male genital tract. AXL is highly enriched in the testes and epididymis, but there is no expression in either the prostates or seminal vesicles of both mice and humans. In addition, the Zika virus disrupts the blood–testis barrier through the polyubiquitination of matrix metallopeptidase 9 (MMP-9), thereby enhancing MMP-9, which degrades the tight junction proteins and type IV collagen of the blood–testis barrier. This facilitates the entry of the virus into the testes.

### 2.4. Lassa Virus and Male Fertility

With the increasing prevalence of Lassa virus disease, there are concerns about its potential to cause transgenerational defects due to its persistence in recovered patients or result in subfertility/infertility among survivors. There is evidence that suggests that the viral antigen perpetuates in the epithelial cells of breasts, theca and stromal cells in ovaries and trophoblastic cells in the placenta and has been associated with some endocrine tissues that might affect fertility in females.

In males, there is evidence that suggests that Lassa virus is excreted from the semen up to three months after infection [45,46], suggesting the possibility of the sexual transmission of the disease among previously infected persons [46]. This occurrence was similarly reported among two patients previously treated with a combination of ribavirin and favipiravir with prolonged detectable viral RNA in the blood and semen, which suggests the possibility of the sexual transmission of Lassa virus after treatment [47]. Indeed, another study reported that viral RNA persists in the semen of patients after viremia has been resolved, and this was accompanied by epididymitis [48]. This suggests that the male reproductive tract is one site of antigen persistence after Lassa virus infection. This noted persistence might be a result of memory T cells formed after an elevated acute response of T cells to Lassa virus [48]. Hence, further studies should be geared towards modulating the role of T cells in Lassa virus disease in order to lessen its systemic and prolonged deleterious effects.

### 2.5. Crimean–Congo Hemorrhagic Fever and Human Reproductive System

Crimean–Congo hemorrhagic fever virus (CCHFV) has not been proven to be present in the human reproductive system but has been identified in the testes and ovaries of a non-primate. A study of three infected monkeys shows that these monkeys had features of unilateral inflammation of the testis and the presence of the Crimean–Congo hemorrhagic fever virus antigen and RNA in these animals. This is the first notable evidence reporting the persistence of this virus in the male genital tract, which suggests the possibility of sexual transmission [49].

A fatal CCHFV case in a 53-year-old man presenting with fever and acute scrotal swelling was diagnosed as epididymo-orchitis. This suggests the possibility of CCHFV causing serious complications in male reproduction. Therefore, further studies are required to elucidate the possible effect of this virus on male reproductive function and its implication on fertility.

### 2.6. Effects of Ebola Virus on Male Fertility

Ebola virus (EBOV) ribonucleic acid (RNA) has been previously detected and noted to persist for up to 290 days among previously hospitalized Ebola patients. In addition, semen from Ebola patients was shown to carry Ebola virus 70 days after the onset of the illness. A similar report on the persistence of Ebola in semen reported that the virus remained in the semen 82 days after a previous Ebola virus disease (EVD) outbreak [50]. The infection of the testes by EBOV probably takes place via viremia during acute Ebola virus disease. The viral antigen was detected in the seminiferous tubules from a certain Ebola virus disease case [51]. In experimental animal models of Ebola virus infection, it was noted that particles of Ebola virus were present within the endothelium, monocytes and interstitial cells of the

testes of the infected animals [52]. Ebola virus can be actively spread for longer periods through its release from the immune response of the testes. Hence, the World Health Organization (WHO) suggests that Ebola virus patients abstain from sex for three months post-recovery or use a condom within this period to prevent the spread of the virus [53]. WHO has equally recommended that if semen has not been tested, safe sexual activity should be engaged in for at least 12 months following the onset of EVD. The Centers for Disease Control (CDC) also advises that contact with semen from patients who recovered from Ebola virus disease must be especially avoided when the semen of such people has not been screened for the presence of the virus [53,54]. There is a need to understand the lifespan of Ebola virus RNA within the semen of male Ebola virus disease survivors so as to guide the decisions necessary to avoid sexual transmission. In addition, further studies are required to understand the relevance of viral RNA detection to Ebola virus infection.

### 2.7. Monkeypox Virus

Monkeypox virus is a re-emerging virus with pockets of outbreaks presently in about 30 countries of the world. There is evidence that suggests its presence and effect on the male reproductive system. It has been previously reported that monkeypox virus can induce direct cytopathic effects on and damage to the testes and other parts of the human reproductive tract, causing interstitial orchitis and seminiferous tubule degeneration [55,56]. There are also growing concerns that monkeypox virus could be sexually transmitted because of its persistence in semen. Antinori et al. [57] reported the presence of the virus in human male semen. Further, Bragazzi and colleagues pointed out a correlation between sexual practices and clinical signs of monkeypox virus, noting that men who have sex with men, practice unsafe sex and have a prior history of sexually transmitted diseases are more susceptible to monkeypox infection. Heskin et al. reported the first case of the possible sexual transmission of monkeypox among patients who had no recent travel history but became infected through sexual contact [58]. Noteworthy are characteristic lesions in the ano-genital areas of monkeypox patients, which further suggest the possibility of sexual transmission [59,60].

Similar to reports on some other viral zoonotic diseases, monkeypox virus can be detected in semen. This could be a result of a high viral load, local or systemic inflammation, the disruption of the blood–testes barrier or the replication of the virus in accessory glands [61,62]. With the presence of monkeypox virus in the semen and male reproductive tract, there is a need to further understand the factors responsible for the high viral load in semen and the possibility of transmitting the virus to female partners and offspring. It is also yet to be discovered whether monkeypox infection can lead to future infertility in these patients. A summary of the effects of viral zoonotic diseases on male reproduction is given in Table 1.

**Table 1.** Viral zoonotic diseases and their effects on male reproduction.

| Viral Zoonotic Disease | Effect on Male Reproduction |
| --- | --- |
| Influenza | Orchitis [19]; Abnormal semen parameters and sperm DNA integrity [17]; Chromosomal aberration in spermatogonia [14,15]; Decreased sperm production [16]; With fever: transient release of abnormal sperm and impaired chromatin structure [12]. |

**Table 1.** *Cont.*

| Viral Zoonotic Disease | Effect on Male Reproduction |
|---|---|
| COVID-19 | Orchitis [24];<br>Germ cell destruction, very low spermatozoa in seminiferous tubules, basement membrane thickening and leukocyte infiltration [24];<br>Orchitis;<br>Impaired seminal parameters, induced orchitis and hypogonadism [29,30]. |
| Zika virus | Detected in semen [35,37];<br>Dysregulation of germ-cell–Sertoli-cell junction signaling [42];<br>Downregulated secretion of inhibin B [43]. |
| Lassa virus | Epididymitis [48];<br>Present in semen with possibility of sexual transmission [46,47]. |
| Crimean–Congo hemorrhagic fever | Epididymo-orchitis [63]. |
| Ebola virus disease | Virus persists in semen long after infection [50];<br>Possibility of sexual transmission [45]. |
| Monkeypox | Virus in semen [59];<br>Possible sexual transmission [58];<br>Orchitis and degeneration of seminiferous tubules [56]. |

## 3. Conclusions

As noted in this review, viral zoonotic diseases could pose a major risk to male fertility. It could be a major contributor to the prevalence of male factor infertility. It is evident that viral zoonotic diseases can disrupt testicular function through inflammatory responses, the impairment of semen DNA and sperm parameters and even linger in semen after recovery from the disease. While some of these viruses are not detectable in the semen, a good number of them persist in the semen long after recovery from the disease. There is a need to fully understand the persistence of some of these viral zoonoses in semen and assess their potential for sexual transmission. In addition, there is a need to assess the effects of the persistence of viral zoonoses on future male fertility and the health of offspring.

**Author Contributions:** Conceptualization, A.F.A., O.L.O. and L.O.A.; literature collection and curation, A.N.O., V.A. and H.O.A.; writing—original draft, A.N.O., V.A. and H.O.A.; writing—review and editing, O.L.O., L.O.A., C.S.N. and A.O.A., final proofreading, J.W.A. and A.F.A. All authors have read and agreed to the published version of the manuscript.

**Funding:** This research did not receive any specific grants from funding agencies in the public, commercial or not-for-profit sectors.

**Institutional Review Board Statement:** Not applicable.

**Informed Consent Statement:** Not applicable.

**Data Availability Statement:** Not applicable here.

**Conflicts of Interest:** The authors declare no conflict of interest.

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
