# Peer review of "Viral Zoonotic Diseases of Public Health Importance and Their Effect on Male Reproduction"

_zoonoticdis, doi:10.3390/zoonoticdis2040023_

Round 1

Reviewer 1 Report

In this review article, the authors give a summary of several viruses whose infections may be involved in male fertility. 

Several suggestions:

1.      Page 2, please add references in the sentence [Other notable zoonotic disease outbreaks include the Ebola-virus outbreak in 2013, Russian flu in 1977, the Spanish flu in 1918, the Middle East Respiratory Syndrome Coronavirus (MERS-CoV) in 2012, Human Immunodeficiency virus in 1981 and Swine flu in 2009. Exposure to zoonotic pathogen disrupts the human physiology with short or long term effect on body organ functions.].

2.      Please use [COVID19] in the entire manuscript, not others, like [Covid19].

3.      Page 3, please delete [and] in [Effect of Viral zoonotic Diseases and on male reproduction].

4.      Please use [Zika] in the entire manuscript, not others, like [zika].

5.      Page 6, please add the full-name first when first mentioning the [EBOV]; [EVD].

6.      Please use [Monkeypox] in the entire manuscript, not others, like [Monkey pox].

Author Response

Response to Reviewer 1 comment

Several suggestions:

  1. Page 2, please add references in the sentence [Other notable zoonotic disease outbreaks include the Ebola-virus outbreak in 2013, Russian flu in 1977, the Spanish flu in 1918, the Middle East Respiratory Syndrome Coronavirus (MERS-CoV) in 2012, Human Immunodeficiency virus in 1981 and Swine flu in 2009. Exposure to zoonotic pathogen disrupts the human physiology with short or long term effect on body organ functions.].

Response:     Thanks, reference provided as [6]

  1. Please use [COVID19] in the entire manuscript, not others, like [Covid19].

Response: Thanks this have been carried out all through the manuscript.

  1. Page 3, please delete [and] in [Effect of Viral zoonotic Diseases and on male reproduction].

Response: thank you for this observation, necessary correction have been effected

  1. Please use [Zika] in the entire manuscript, not others, like [zika].

Response: this suggestion has been effected, thank you.

  1. Page 6, please add the full-name first when first mentioning the [EBOV]; [EVD].

                Response: necessary corrections have been made, thank you.

6. Please use [Monkeypox] in the entire manuscript, not others, like [Monkey pox].

Response: thank you for this suggestion; I have done the needed corrections.

Reviewer 2 Report

Dear Authors

After reviewing your article, I must point out to you that:

1.- You say that your article is a review. However, you haven't indicated, anything that refers to your paper being a narrative review. 

If you have developed a search structure, even if it is because you have done a scoping review, you should include the Material and methods section.  It is therefore necessary to include this information at some point in the work because the time frame of your search is not even indicated. 

Author Response

Response to Reviewer 2 comment

1.- You say that your article is a review. However, you haven't indicated, anything that refers to your paper being a narrative review. 

If you have developed a search structure, even if it is because you have done a scoping review, you should include the Material and methods section.  It is therefore necessary to include this information at some point in the work because the time frame of your search is not even indicated. 

Response: thank you for your observation this is the response provided

Search method

Research based databases like google scholar, PubMed and web of science was used for this review. Each of the virus included in this study were reviewed by following a keyword format which juxtaposed the virus with (male fertility or testes or semen parameters or sperm parameters). For instance, the keyword syntax search for Coronavirus was (((COVID-19) OR (SARS-CoV-2)) OR (Coronavirus)) AND ((((Male fertility) OR (Testes)) OR (Semen parameters)) OR (Sperm parameters)). Articles included in this study are original research articles on human and other mammals while research on other animals and those not reported in English were excluded. Major attention was paid to articles published between 1990 till date.